# Efficacy, safety, and immunogenicity of Lassa fever vaccines: A living systematic review and landscape analysis of vaccine candidates

Ariel Bardach[1,2]*, Mabel Berrueta[3], Agustín Ciapponi[1,2], Juan M. Sambade[4], Noelia Castellana[4], Jamile Ballivian[4], Martín Brizuela[4], Julieta Caravario[4], Daniel Comande[4], Esteban Couto[4], Agustina Mazzoni[4], Vanesa Ortega[4], Edward P. K. Parker[5], Florencia Salva[4], Katharina Stegelmann[4], John S. Schieffelin[6], Xu Xiong[6], Andy Stergachis[7], Flor M. Munoz[8‡], Pierre Buekens[6‡]

1 Argentine Cochrane Center, Institute for Clinical Effectiveness and Health Policy, Buenos Aires, Argentina, 2 Center for Research in Epidemiology and Public Health, CONICET, Buenos Aires, Argentina, 3 Department of Mother and Child Health, Institute for Clinical Effectiveness and Health Policy, Buenos Aires, Argentina, 4 Institute for Clinical Effectiveness and Health Policy, Buenos Aires, Argentina, 5 Department for Infectious Disease Epidemiology and International Health, London School of Hygiene and Tropical Medicine, London, United Kingdom, 6 Department of Epidemiology, Celia Scott Weatherhead School of Public Health and Tropical Medicine, Tulane University, New Orleans, Louisiana, United States of America, 7 University of Washington Schools of Pharmacy and Public Health, Seattle, Washington, United States of America, 8 Departments of Pediatrics and Molecular Virology and Microbiology, Baylor College of Medicine, Houston, Texas, United States of America

‡ PB and FMM are co-senior authors
* abardach@iecs.org.ar

## Abstract

### Background

Lassa fever (LF) is an acute viral hemorrhagic illness endemic in West Africa, representing significant public health challenges, particularly for pregnant persons and children who experience higher morbidity and mortality. Although several vaccine candidates are being developed, no LF vaccine has been licensed yet.

### Methods

We conducted a living systematic review (LSR) of the literature to evaluate the safety, efficacy, effectiveness, and immunogenicity of LF vaccines. We performed biweekly searches in major biomedical databases, trial registries, preprint servers, and other sources. Eligible studies included preclinical studies, clinical trials, and observational studies published from January 2014 to April 2025. Reviewer pairs screened studies extracted data (REDCap), and assessed risk of bias independently. Data synthesis involved random-effects pairwise and proportion meta-analyses (R software), with GRADE assessment of evidence certainty. PROSPERO registries: (CRD42024514513; CRD42024516754).

**Data availability statement:** All relevant data are within the manuscript and its Supporting Information files.

**Funding:** This work was supported by the Safety Platform for Emergency Vaccines (SPEAC), a Brighton Collaboration project funded by the Coalition for Epidemic Preparedness Innovations (CEPI). The funder (CEPI) had no role in study design, data collection and analysis, decision to publish, or preparation of the manuscript. The views expressed are those of the authors and do not necessarily reflect the positions of their institutions or the funder.

**Competing interests:** The authors have declared that no competing interests exist. FMM reports research funding from Pfizer, and consulting fees from Sanofi and MSD, unrelated to this work.

## Results

Searches retrieved 1423 records, including 51 studies, 2 clinical trials in adults involving 88 vaccinated persons, and 49 preclinical studies of 30 vaccine candidates. Trials evaluated Recombinant Measles-Vectored (MV-LASV) and Recombinant Vesicular Stomatitis Virus-based (rVSVΔG-LASV-GPC) LF vaccine candidates. No published clinical trials were found to evaluate LF vaccines in special populations such as pregnant persons, infants, children, or adolescents. Although injection site reactogenicity was reported, no vaccine-related serious adverse events (SAEs) were reported in study participants. Immunogenicity was robust in adults, with vaccines achieving around 95% seroconversion at 30 days. Preclinical data evaluated nine different platforms. Findings are disseminated via an interactive online dashboard (https://safeinpregnancy.org/living-systematic-review-lassa/).

## Conclusion

Currently, two LF vaccine candidates that have advanced to clinical trials exhibit high immunogenicity, but the safety profile in healthy adults is still limited. Clinical evidence in pregnant persons, infants, children, and adolescents is absent. Vaccine platforms of interest have been identified in preclinical studies, providing information on those that could advance to clinical studies.

## Introduction

Lassa fever (LF) is an acute viral hemorrhagic illness caused by Lassa virus (LASV), an *Mammarenavirus* endemic in several West African countries [1,2]. While primarily zoonosis is transmitted via contact with infected *Mastomys* rodents or their excreta, human-to-human transmission can also occur through contact with bodily fluids of infected individuals [3,4]. LASV causes recurrent annual outbreaks within the endemic "Lassa fever belt" in West Africa (mainly Nigeria, Sierra Leone, Liberia, and Guinea) [3,5]. Lassa virus exhibits significant genetic diversity across four distinct phylogenetic lineages, with lineages I-III predominantly circulating in Nigeria and lineage IV endemic to West African [1].

LF represents a substantial public health burden in endemic regions. Estimates suggest 100,000–300,000 infections and 5,000–10,000 deaths occur annually, although these figures likely underestimate the burden due to surveillance and diagnostic challenges [1,6,7]. While many infections are mild or asymptomatic (80%), about 20% progress to severe disease with multi-organ involvement [1,8]. The overall case fatality rate (CFR) is estimated at 1–2%, but it can be significantly higher (≥15%) among hospitalized patients and during outbreaks [9,10]. Populations at high-risk include pregnant women, with reported maternal CFRs reaching 34% and fetal loss rates exceeding 60–80% [11,12], and children, also showing significant morbidity and mortality compared with adults, though data remain limited [5,13,14]. Sensorineural hearing loss is a common sequela, affecting up to one-third of survivors [15].

The Josiah strain serves as the prototypical representative of lineage IV and has been extensively characterized in laboratory studies [16]. The majority of vaccine candidates under preclinical and clinical investigation have predominantly utilized the glycoprotein (GP) from the Josiah strain of lineage IV as the primary immunogen, with homologous lineage IV strains employed in challenge studies [1,16].

No vaccine against LF is currently licensed for human use [16]. The virus's genetic heterogeneity, the lack of established immune correlates of protection, and logistical challenges for conducting large efficacy trials in resource-limited settings with sporadic outbreaks have hampered vaccine development [1,17]. However, renewed focus, driven by recurrent large outbreaks, recognition by the World Health Organization (WHO) of LF virus as a priority pathogen with epidemic potential, and initiatives like the Coalition for Epidemic Preparedness Innovations (CEPI) seeking to support preventive strategies, have spurred the development of numerous vaccine candidates [16–18].

Given the urgent need for effective LF prevention and the evolving vaccine pipeline, we conducted a living systematic review – a dynamic method to continuously and rapidly incorporate new evidence from emerging studies – to explicitly and continuously evaluate the safety, immunogenicity, efficacy, and effectiveness of LF vaccine candidates across study designs, and to describe the evolving vaccine pipeline for decision-makers. Although our research hub focuses on pregnant persons, children, and adolescents [19,20], for this condition, we are capturing all available, relevant data from adult populations to collect indirect evidence and preclinical studies to understand the current vaccine development pipeline.

## Methods

This LSR was conducted following Cochrane [21] and WHO guidance and reported following PRISMA recommendations and extensions for living systematic reviews [22,23]. The full methodology is available in our published protocol [24] and the PROSPERO registries (CRD42024514513; CRD42024516754).

Briefly, we included (i) clinical studies (randomized, non-randomized, observational) of any LF vaccine candidate; (ii) preclinical studies in animal models reporting immunogenicity and/or protection; (iii) all languages and publication statuses (including preprints and conference abstracts). We excluded narrative reviews, editorials, modeling without primary vaccine data, studies on non-Lassa arenaviruses, insufficient data for outcomes of interest, or lack of original data. There were no restrictions related to dosing, administration schedule, comparators, or control groups. We also searched historical reviews for candidate vaccines.

The primary outcomes were safety, efficacy, effectiveness, and immunogenicity of vaccines administered primarily to adults and high-risk populations such as pregnant persons and those <18 years of age. Efficacy/effectiveness outcomes included clinical Lassa outcomes. Safety assessments included reactogenicity, serious adverse events (SAEs), adverse events of special interest (AESI) and obstetric and neonatal outcomes (when available) conforming with standardized definitions from the Global Alignment of Immunization safety Assessment in pregnancy (GAIA) [25] and Safety Platform for Emergency vACcines (SPEAC) [26] initiatives where applicable. Immunogenicity was measured through serologic markers, including LASV-specific IgG and IgM antibody geometric mean titers (GMT), seroconversion, and neutralizing antibody levels [27]. Secondary variables included platform, dose, schedule, setting, population, funding, and follow-up. For trials lacking explicit seroconversion definitions, we operationalized seroconversion as a prespecified increase from baseline above study-defined thresholds for binding or neutralizing antibodies.

We conducted biweekly searches across multiple databases, clinical trial registries, and preprint servers. The selected timeframe spanned from January 2014 to April 2025. No restrictions were applied regarding language or publication status. We searched MEDLINE/PubMed, Embase, Cochrane CENTRAL, Scopus, Web of Science, ClinicalTrials.gov, EudraCT/EU-CTR, ISRCTN, PACTR, WHO ICTRP, and preprint servers (medRxiv, bioRxiv). All sources were last searched in April 2025, with automated biweekly alerts thereafter. Full strategies are in S1 File. Study selection (titles/abstracts and full texts), data extraction (via REDCap) [28], and risk of bias assessments according to study design, using RoB2 for RCTs and the National Institutes of Health (NIH) tools for non-randomized studies were performed independently

and in duplicate by a pair of reviewers [29,30]. Disagreements were resolved by consensus, and exclusions were documented with reasons (S1 Table 1 in S1 File). Certainty of evidence for comparative outcomes was assessed using the Grading of Recommendations Assessment, Development and Evaluation (GRADE) framework [31].

Where possible, we conducted random-effects meta-analyses using R software version 4.2.2 [32], including pairwise and proportion analyses [33]. We used DerSimonian–Laird random-effects models for pairwise and proportion meta-analyses when ≥2 comparable studies were available. Other details on statistical methods are referred to in the full protocol [24]. Preclinical studies were not meta-analyzed; however, they were systematically reviewed to assess the current vaccine development pipeline, including which candidates remain active and which have been discontinued, as well as ongoing studies. Statistical heterogeneity was analyzed by subgroups, when it was possible. An I2 value greater than 60–70% indicated substantial heterogeneity, while a value below 30% indicated a low level of heterogeneity. We planned subgroup/sensitivity analyses by platform, dose, and time window when data allowed, and did not pool when conceptual/statistical heterogeneity was high. For missing data, we contacted authors when feasible; otherwise, we analyzed available data without imputation. Outcome denominators reflect the number with non-missing data for each endpoint. Regarding ethics in included animal studies: Although we did not perform animal research, when reported we extracted statements on institutional approval and animal welfare compliance for preclinical studies.

A flow diagram detailing the study selection process is also available on the interactive dashboard (https://safeinpregnancy.org/living-systematic-review-lassa/).

## Results

The searches retrieved 1423 unique records across all databases and sources up to April 2025. Following title/abstract screening and full-text review according to the Study Selection Flow Diagram (Fig 1), we included 51 studies in the current synthesis: two clinical trials and 49 preclinical studies. The two clinical trials enrolled a total of 88 vaccinated adults. Additionally, five ongoing clinical trials were reviewed and incorporated into the landscape analysis. Search strategy is listed in S1 File, and the excluded studies and reason for exclusion are shown in S1 Table 1 in S1 File. For immunogenicity outcomes, 'seroconversion' was standardized as a post-vaccination increase meeting each trial's predefined threshold for anti-GPC IgG or neutralizing antibodies. Safety data are summarized in a consolidated adverse-event table (see S1 Table 5 in S1 File), stratified by candidate and time window, which complements the detailed forest plots in the Supplement.

### Characteristics of included studies

The two clinical studies included were phase I placebo-controlled randomized clinical trials (RCT) assessing LF vaccine candidates in healthy adults aged between 18 and 70 years. Participants were recruited from the United States (US) and Belgium. The vaccine candidates evaluated comprised a recombinant vesicular stomatitis virus vector vaccine (rVSVΔG-LASV-GPC) [34] and a measles-vector-based vaccine candidate (MV-LASV) [35]. Both studies assessed vaccine safety, reactogenicity, and immunogenicity, employing varying dosing schedules and follow-up periods. Detailed characteristics of these clinical studies are summarized in Table 1.

The 49 preclinical studies evaluated various vaccine candidates including those that progressed to phase I clinical trials, covering multiple vaccine platforms, including other recombinant viral vector, inactivated virus, live attenuated virus, self-assembled vaccine (SAV), viral like particles (VLP), virus replicon particles (VRP) and mRNA, which were tested in different animal models. These studies primarily reported immunogenicity and protective efficacy against LASV challenge.

### Clinical studies in target populations

No clinical trials or observational studies of LF vaccines reported including pregnant individuals or unintended pregnancies, infants, children, and adolescents. See section Vaccines in Clinical Development.

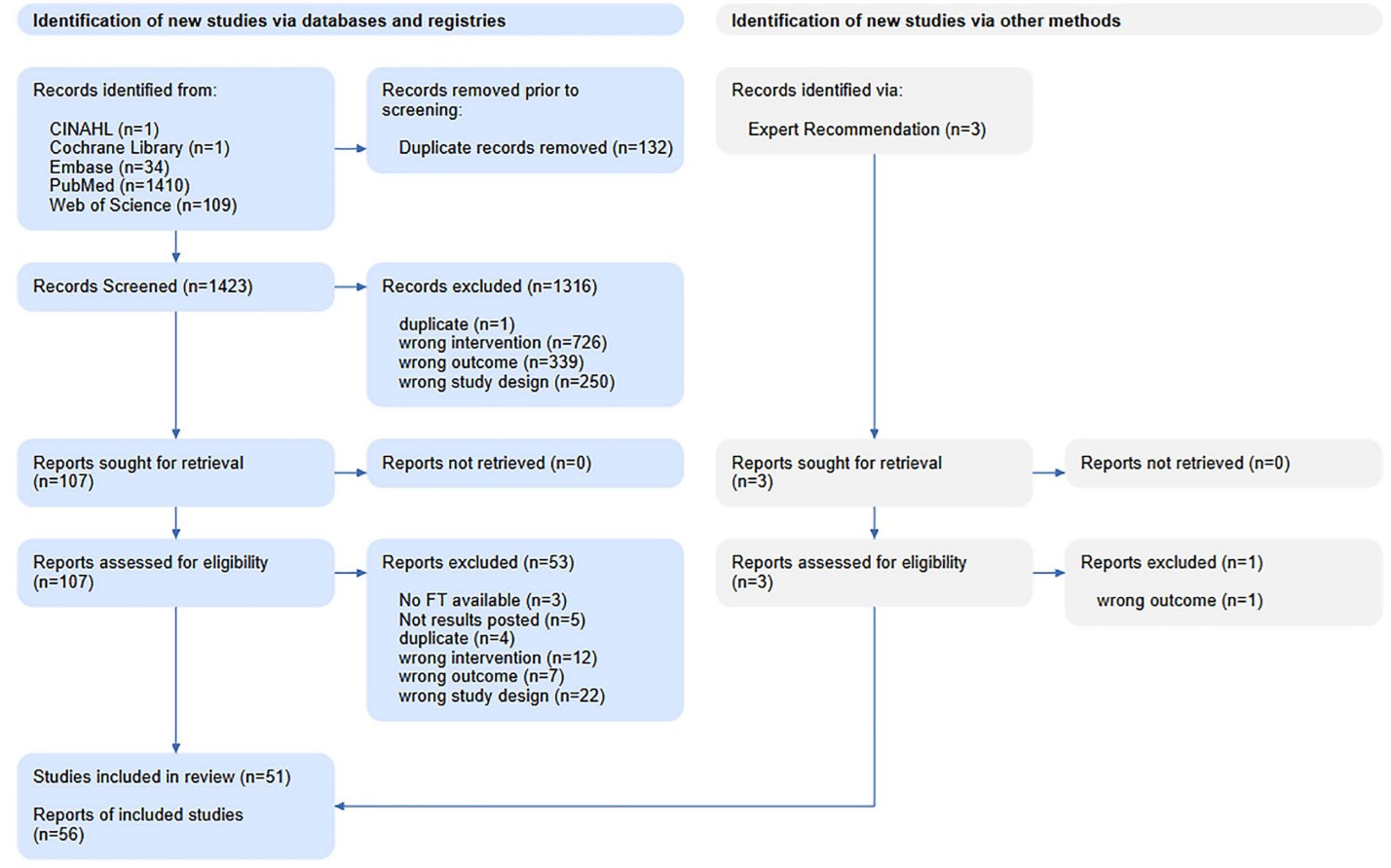

**Fig 1. Study Selection Flow Diagram.**

## Safety outcomes in adults

Across adult clinical trials of attenuated recombinant LF vaccines (including MV-LASV and rVSVΔG-LASV-GPC) administered as intramuscular (IM) injection, solicited local and systemic adverse events were frequently observed but were almost uniformly mild to moderate in intensity and resolved without intervention within 14 days after vaccination. No vaccine-related SAEs—such as hospitalization, death, seizures, or anaphylaxis— were reported. Nor were any pre-specified AESIs reported, although one case of hypoacusis was observed in the high-dose arm ($1 \times 10^5$) of the rVSVΔ trial [36]. Unsolicited adverse events occurred at similar rates in vaccine and placebo groups, and a pooled meta-analysis found no significant increase in either frequency or severity of unsolicited adverse events among vaccine recipients. Overall, the safety profile in adults supports continued clinical development of these attenuated recombinant candidates.

Arthralgia occurred at a rate of 32.02 events per 100 participants (95% CI: 12.48–54.82) within 0–28 days post-vaccination with the rVSVΔ vaccine ($I^2 = 47\%$), and 2.71 events per 100 participants (95% CI: 0.00–16.45) up to day 56 with the MV-LASV vaccine ($I^2 = 61\%$). The incidence of fever within 30 days of rVSVΔ vaccination was 31.49 per 100 participants (95% CI: 10.73–56.24, $I^2 = 43\%$), and 4.16 per 100 participants (95% CI: 0–12.60, $I^2 = 0\%$) up to day 56 with MV-LASV vaccines, although a wide range across dosages was observed.

Diarrhea was reported at 11.85 events per 100 participants (95% CI: 1.44–27.60) within 28 days post-vaccination with rVSVΔ vaccine ($I^2 = 32\%$), and 18.47 per 100 participants (95% CI: 8.31–31.13) up to day 56 with MV-LASV vaccines ($I^2 = 0\%$).

**Table 1. Characteristics of Included Clinical Studies.**

| Author and year | Vaccine Candidate (Platform) | Study Phase | Population N (Vaccine/ Placebo) | Dosage– plaque forming unit (pfu) | Dose number (interval) | Age | Country | Key Outcomes Reported | Risk of bias |
|---|---|---|---|---|---|---|---|---|---|
| Tschisma-rov et al., 2023 [36] | MV-LASV (Recombinant Measles-Vectored) | Phase 1 | 60 (48/12) | $2 \times 10^4$ $1 \times 10^5$ | 2 (28 days) | 18–55 years | Belgium | The trial evaluated two exposure arms with different dose levels of the MV-LASV vaccine. Both arms were well tolerated and elicited binding and neutralizing antibody responses by Day 29, which increased through Day 84 and remained stable through Day 252. No serious adverse events were reported. | Some concerns* |
| Malkin et al., 2023 [37] | rVSVΔG-LASV-GPC (Recombinant Vesicular Stomatitis Virus-based) | Phase 1 | 52 (40/12) | $2 \times 10^4$ $2 \times 10^5$ $2 \times 10^6$ $2 \times 10^7$ | 1 dose or 2 doses for $2 \times 10^7$ (6–20 weeks) | 18–50 years | USA | The rVSVΔG-LASV-GPC vaccine was well tolerated and immuno-genic across a wide dose range. Binding and neutralizing antibodies were present by Day 29, increased through Day 169, and persisted through Day 252. No related serious adverse events were reported. | High* |

*Both clinical trials were RCTs and were assessed using the Cochrane RoB tool. Malkin's study was judged to have a low risk of bias across most domains, except for selection of the reported results, with overall high risk of bias. Tschismarov's paper was judged to have some concerns for risk of bias, mainly due to concerns in the randomization process and missing outcome data domains. Detailed risk of bias assessments for each study are available in the Supplementary Material (S1 Table 2 in S1 File) and on the interactive dashboard.

For myalgia, the incidence within 28 days of rVSVΔ vaccination was 56.22 per 100 participants (95% CI: 20.78–88.82, $I^2 = 83\%$), and 24.99 per 100 participants (95% CI: 13.38–38.57, $I^2 = 0\%$) between days 31 and 180 with MV-LASV vaccines.

Additional adverse events reported with both vaccines included chills, fatigue, headache, injection site reactions, malaise, presyncope and syncope, rash, and vomiting. Meta-analyses and forest plots are presented in S1 Fig A-Dfor arthralgia, diarrhea, fever and myalgia, with full details available at: https://safeinpregnancy.org/comparative-meta-analyses-lassa-fever

## Immunogenicity outcomes in adults

Malkin et al. reported data within 30 days post-vaccination that was included in the meta-analysis, encompassing a total of 39 vaccine recipients from the US who received the rVSVΔ vaccine at varying dosages (ranging from $2x10^4$ to $2x10^7$ pfu). Across these four dosage subgroups, the proportion of patients experiencing seroconversion—though not explicitly defined in the publication—ranged from 75% (at $2x10^4$ pfu) to 100% (at $2x10^6$ and $2x10^7$ pfu), with anti-GPC IgG antibodies used as the immunological marker. The overall pooled proportion was 95.16% (95% CI: 79.68–100.00), with moderate heterogeneity ($I^2 = 40\%$, p = 0.17). Fig 2.

MV-LASV induced substantial levels of LASV–specific IgG antibodies at both dose levels. Although no dichotomous immunogenicity outcome (such as seroconversion) was defined for this vaccine, immunogenicity was assessed at multiple timepoints (days 0, 14, 28, 42, 56, 182, and 365). Peak GMTs were observed on day 42: 62.9 EU/mL (95% CI: 38.2–103.8) in the low-dose group and 145.9 EU/mL (95% CI: 87.4–243.8) in the high-dose group. Titers remained significantly elevated through day 56 in both groups and persisted through day 182 in the high-dose group [36].

## Vaccines in clinical development

Several first-generation vaccine candidates have been discontinued or have failed to progress beyond early-phase trials, primarily due to strategic reprioritization, limited funding, or lack of supportive immunogenicity data. Three others remain

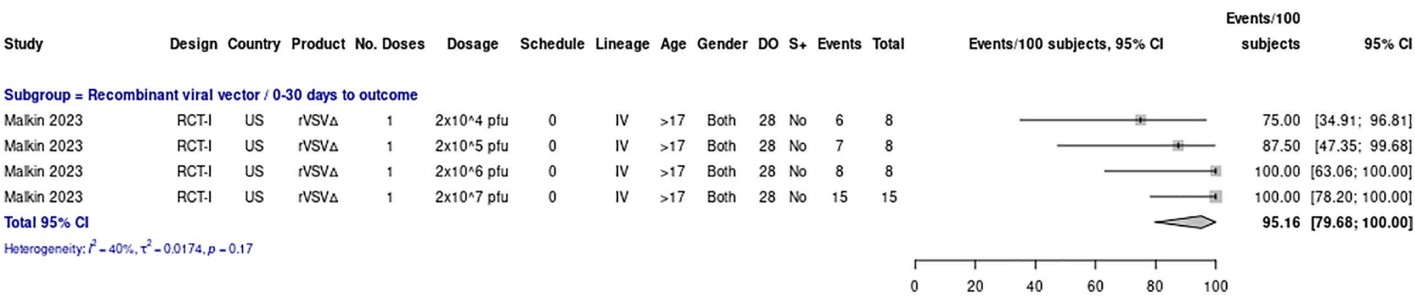

**Fig 2. Seroconversion rate across different dosages of rVSVΔG-LASV-GPC vaccine (30-day outcomes).**

active in the pipeline, with one candidate undergoing Phase II evaluation in endemic regions. The summary of the current clinical landscape, distinguishing between active, paused, and discontinued candidates based on trial registry status, sponsor communications, and updated funding from CEPI and partner institutions is shown in Table 2. We identified six vaccine candidates that reached Phase I or higher, as detailed in the same table Fig 3.

A Phase II trial of the rVSVΔG-LASV-GPC candidate, registered at clinicaltrials.org under NCT05868733, including children aged 2–17 years from West Africa (Ghana, Liberia, and Nigeria), is recruiting participants, with an estimated primary completion date in December 2026 [44]. Another phase I trial under NCT06546709 is being carried out in Maryland (US) with adults 18–50 years old to receive the experimental LASSARAB vaccine, with an estimated date of completion in March 2026 [45]. Recently, a protocol for a phase 1 trial sponsored by CEPI and the University of Oxford to evaluate the safety and immunogenicity of ChAdOx1 Lassa vaccine in healthy adults between 18 and 55 years in the UK was published, and the estimated date for completion is September 2027 [43]. No data area is yet available from these trials to include in the LSR. This updated analysis of the vaccine pipeline is crucial for prioritizing future research investments and highlights the need for coordinated global efforts to support the clinical evaluation of LASV vaccines, particularly those targeting special populations.

**Table 2. Status of Lassa Fever Vaccine Candidates in Clinical Development.**

| Platform | Candidate (Developer) | Phase | Year | Population | N | Trial registration (status) |
|---|---|---|---|---|---|---|
| Recombinant viral vector | rVSVΔG-LASV-GPC (IAVI/ CEPI/ PHAC) | 2b | 2022–2023 | 18 months– 70 years (incl. PLWH) | 612 | NCT05868733/ PACTR202210840719552 [38,39] (Ongoing) |
| Recombinant viral vector | rVSVΔG-LASV-GPC (IAVI/ CEPI/ PHAC) | 1 | 2021 | 18–50 years | 110 | NCT04794218/ PACTR202106625781067. [37] (Completed) |
| Recombinant viral vector | EBS-LASV (Emergent BioSolutions) | 1 | 2021 | 18–50 years | 108 | PACTR202108781239363 [40] (Discontinued) |
| Recombinant viral vector | MV-LASV (Themis/ BioNTech) | 1 | 2019 | 18–55 years | 60 | NCT04055454 [36] (Discontinued) |
| DNA | INO-4500 (Inovio/ CEPI) | 1 | 2021 | 18–50 years | 220 | NCT04093076 [41] (Discontinued) |
| DNA | INO-4500 (Inovio/ CEPI) | 1 | 2019 | 18–50 years | 60 | NCT03805984 [41] (Discontinued) |
| Inactivated viral vector | LASSARAB | 1 | 2025 | 18–50 years | 55 | NCT06546709 [42] (Ongoing) |
| Recombinant viral vector | ChAdOx1 LassaJ | 1 | 2024 | 18-55 years | 31 | ISRCTN16084957 [43] (Ongoing) |

See expanded table in the Supplementary Material (S1 Table 4 in S1 File).

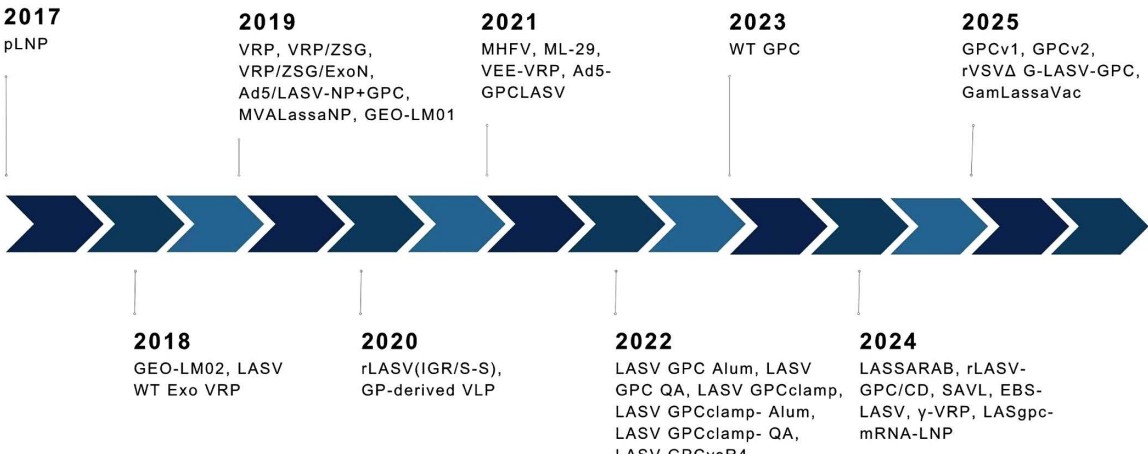

**Fig 3. Timeline of platforms evaluated in preclinical studies; dates denote publication of results unless otherwise indicated.**

## Vaccines in preclinical development

Among the 49 published articles reporting preclinical studies, 51% (n = 25) were published between 2020 and 2025, indicating renewed interest in these vaccines. Most studies were conducted in the United States (n = 31), followed by France (n = 3) and China (n = 3). The primary animal models were rodents (73%, n = 33) and non-human primates (NHP; 33%, n = 15). Rodent models included mice and guinea pigs; notably, immunocompetent mice are not susceptible to LASV.

Thirty candidates were identified to be in preclinical development in the search, employing diverse platforms such as viral vectors (e.g., measles virus–based MV-LASV, vesicular stomatitis virus–based rVSVΔG-LASV-GPC), DNA vaccines (e.g., INO-4500), mRNA vaccines, recombinant proteins, and live-attenuated viruses [18,46], see Table 3 and Fig 3. The most frequently evaluated platforms were recombinant viral vectors (n = 17), live-attenuated (n = 6), and protein subunits (n = 3). Reported outcomes focused on immunogenicity, efficacy, and safety. 22 candidates remain active as of 2025. Several earlier ones (e.g., recombinant adenovirus constructs from the early 2010s) show no recent updates and are inactive. ML-29, a reassorting Mopeia-Lassa virus clone, and LASSARAB, a rabies virus-vectored vaccine platform, are among the most extensively evaluated in preclinical settings and may be prioritized for future clinical development. See Table 3 for details.

Recent preclinical studies demonstrate that multiple platforms—including DNA vaccines such as INO-4500, live-attenuated and reassortant Mopeia-Lassa constructs, recombinant measles- and VSV-vectored candidates, and nanoparticle or polymersome-based formulations—consistently induce robust immunogenicity and confer considerable protection in rodent and non-human primate models. [82–96] These investigations also provide supportive safety data, including biodistribution and toxicology profiles, and show rapid, durable, and cross-lineage protection after single-dose regimens.

## Discussion

This LSR shows that LF vaccine development remains in its early stages, although research and candidate production have increased considerably in recent years, with multiple platforms currently being evaluated in both pre-clinical and clinical trials. Evidence on safety, efficacy, effectiveness, and immunogenicity from clinical studies, however, remains limited. Notably, high-risk populations such as pregnant persons, infants, children, and adolescents have not yet been included in clinical studies. A limited number of platforms are being evaluated for at-risk populations. To date, only one phase 2 trial has been registered that gradually expands eligibility criteria to include children as young as 18 months, adults up

**Table 3. Status of Lassa Fever Vaccine Candidates in Pre-Clinical Development.**

| Platform | Vaccine name(s) | Number of publications | Animals | | Endpoint | | | Lastest publication (Country – year) |
|---|---|---|---|---|---|---|---|---|
| | | | Rodents | NHP | Immunogenicity | Efficacy | Safety | |
| DNA | MHFV[1] | 1 [47] | ✓ | ✓ | ✓ | | | Jiang, J (US-2021) [47] |
| | pLNP | 1 [48] | ✓ | | ✓ | ✓ | | Li, Q (China-2017) [48] |
| Inactivated viral vector | LASSARAB | 2 [49,50] | ✓ | ✓ | ✓ | ✓ | | Scher, G (US-2024) [49] |
| Live attenuated | rLASV-GPC/CD | 2 [51,52] | ✓ | | ✓ | ✓ | ✓ | Carey B (US-2024) [52] |
| | ML-29 | 3 [53–55] | ✓ | | | ✓ | ✓ | Johnson, D (US-2021) [53] |
| | rLASV(IGR/S-S) | 1 [56] | ✓ | | ✓ | ✓ | | Cai, Y (US-2020) [56] |
| Protein subunit | GPCv1, GPCv2 | 1 [57] | ✓ | | ✓ | ✓ | | Wang S (China-2025) [57] |
| | γ-VRP | 1 [58] | ✓ | | ✓ | | | Gorman, J (US-2024) [58] |
| | LASV GPCclamp, LASV GPC QA and others[2] | 1 [59] | ✓ | | ✓ | | | Young, A(Australia-2022) [59] |
| SAV | SAVL[3] | 1 [60] | ✓ | | ✓ | ✓ | | Leblanc, P (US-2014) [60] |
| VLP | GP-derived VLP | 1 [61] | ✓ | | | | ✓ | Muller, H (Germany-2020) [61] |
| VRP | VRP, VRP/ZSG, γ-VRP, VRP/ZSG/ExoN | 1 [62] | ✓ | | ✓ | ✓ | | Kainulainen, M (US – 2019) [62] |
| Recombinant viral vector | rVSVΔ G-LASV-GPC | 6 [63–69] | ✓ | ✓ | ✓ | ✓ | ✓ | Cooper, C (2025) [63] |
| | GamLassaVac | 1 [70] | ✓ | | ✓ | ✓ | ✓ | Popova, O (2025) [70] |
| | ChAdOx1/Padovax | 2 [71,72] | ✓ | | ✓ | ✓ | | Flaxman, A (2024) [72] |
| | Ad5-GPCLASV | 1 [73] | ✓ | | ✓ | | | Wang, M (2021) [73] |
| | LASSARAB, VEE-VRP | 1 [74] | | ✓ | ✓ | | | Kurup, D (2021) [74] |
| | Ad5/LASV-NP+GPC | 1 [75] | ✓ | | ✓ | | ✓ | Maruyama, J (2019) [75] |
| | MVALassaNP | 1 [76] | ✓ | | ✓ | | ✓ | Kennedy, E (2019) [76] |
| | GEO-LM01 | 1 [77] | ✓ | | ✓ | ✓ | | Salvato, M.S (2019) [77] |
| | GEO-LM02 | 1 [78] | ✓ | | ✓ | | | Guzman, C (2018) [78] |
| | LASV WT Exo VRP | 1 [79] | ✓ | | ✓ | | ✓ | Kainulainen, M (2018) [79] |
| RNA | LASgpc-mRNA-LNP | 1 [80] | ✓ | | ✓ | ✓ | | Hashizume, M (2024) [80] |
| | WT GPC | 1 [81] | ✓ | | ✓ | | ✓ | Ronk, A (2023) [81] |

1 Multivalent Hemorrhagic Fever Virus.

2 LASV GPC Alum, LASV GPCclamp- Alum, LASV GPCclamp-QA and LASV GPCysR4.

3 Self-assembled vaccine formulated for Lassa Fever Virus.

* All candidates were deemed to be under active development based on the following criteria: the existence of registered clinical trials, the year of their execution, the publication of relevant studies in scientific databases within the last 3–5 years, and verification on official websites to determine whether the manufacturer has reported updates regarding the vaccine's continuation or discontinuation. See discontinued vaccine studies in the Supplementary Material (S1 Table 3 in S1 File).

to 70 years, and people living with HIV (under defined conditions), although pregnant persons continue to be excluded. However, the registered trial represents an encouraging advancement towards addressing this critical gap [97]. Efficacy was not assessed in phase 2 trials. Interpretation of the current clinical evidence requires caution: sample sizes are small, follow-up is limited, and correlates of protection for LF are not established. As such, antibody responses should be viewed as provisional surrogates. Notably, persistent exclusion of pregnant persons, infants, children, and adolescents raises important equity and ethics concerns; addressing regulatory hesitancy and capacity constraints in endemic regions will be essential for inclusive development.

Despite these limitations, the synthesis of preclinical and early-phase clinical studies and findings enables mapping of vaccine candidates and their progress in development.

The current public health landscape provides a strong rationale for establishing this LSR. LF remains a significant and recurring threat in West Africa, possessing pandemic potential, and the development of safe and effective vaccines is a global health priority [1,17]. Notably, most vaccines in the clinical development stage have focused on a single Lassa virus lineage—typically Lineage IV—despite the known genetic diversity of LASV across affected regions, which may have implications for cross-protection and long-term efficacy. Differences in pathogenicity between lineages impact the development of medical countermeasures. Strain Josiah (lineage IV) is the prototypic strain and has been used in many pre-clinical studies. The vaccines developed targeting this strain could exhibit reduced efficacy against other circulating strains.

It is encouraging that several vaccine candidates in the pipeline are progressing into the clinical phase, primarily in early-phase trials with healthy adults [16]. To date, the main platforms explored include recombinant viral vectors, DNA-based vaccines, and other viral vector approaches. Recombinant platforms, particularly those using rVSVΔG-LASV-GPC, based on a single-dose regimen, showed 30-day seroconversion in approximately 95% of vaccinated individuals, with a safety profile consistent with the results of preclinical studies, progressing to a Phase 2b trial in endemic regions. In contrast, DNA-based vaccine candidates and other viral vectors, such as the MV-LASV platform, have been discontinued or remain at a preclinical stage, indicating limited continuity in their development.

A considerable number of vaccine platforms have undergone preclinical evaluation using rodent and non-human primate models, though several have since been discontinued. Based on the volume of published studies, ML29 and LASSARAB emerge as platforms of particular interest for further development.

Living systematic reviews (LSRs) are accessible, continuously updated syntheses across study designs that support surveillance and evidence-based decisions in this evolving field [19]. Our LSR will rapidly add and disseminate new findings—including trials in children, adolescents, and pregnant women—as they appear. This living approach keeps evidence current for stakeholders [98], unlike traditional reviews that quickly become outdated, especially for vaccines against emerging infections.

Strengths of this LSR include its prospectively registered and comprehensive protocol, adherence to rigorous international standards (Cochrane, PRISMA, GRADE), broad inclusion criteria encompassing diverse study designs, pre-specification of standardized outcome definitions (GAIA, SPEAC, WHO), commitment to frequent updates, and specific focus on addressing evidence gaps in at-risk populations – a key equity consideration in vaccine research [99]. Furthermore, the interactive online dashboard is a powerful tool for rapid knowledge translation, aiming to make complex synthesized evidence accessible and usable for a diverse audience, including researchers, policymakers, clinicians, and public health officials in LF-endemic regions and globally.

The primary limitation of this LSR is the scarcity of published clinical data. Once data emerge, potential limitations common to meta-analyses may arise, including heterogeneity across studies due to differences in vaccine platforms, dosing regimens, populations enrolled (e.g., varying baseline seroprevalence), outcome definitions and measurement methods, and overall study quality. Our pre-specified subgroup and sensitivity analyses are designed to explore and, where possible, explain such heterogeneity. Additionally, reliance on immunogenicity markers as surrogates for clinical protection will likely be necessary in the initial stages, given the challenges in conducting large-scale efficacy trials for LF due to its unpredictable epidemiology and logistical hurdles [35]. The validity of these surrogates remains to be fully established. To date, early clinical data come from non-endemic settings (US and Belgium), which may not fully reflect vaccine performance or safety in populations living in endemic areas with different nutritional status, co-infections, and potential prior LASV exposure.

The implications of this ongoing work are considerable. Firstly, it formally documents and continuously monitors the scientific evidence highlighting the critical need for the ethical and timely inclusion of pregnant persons, children, and adolescents in LASV vaccine clinical development programs. Guidance exists to support such inclusion [99]. Secondly, this LSR provides a robust, transparent, and continually updated evidence synthesis framework that will be indispensable

for informing future clinical trial designs, regulatory submissions, vaccination policy recommendations (e.g., by National Immunization Technical Advisory Groups –NITAGs), and clinical practice guidelines as data accumulate. By methodically assessing evidence across various vaccine platforms, formulations, and populations, this review aims to gather information on the most suitable options and implementation approaches for use in varied environments in near real time, for example, epidemics in endemic areas or the risk of establishment of an outbreak of human transmission through imported cases. With increasing international travel and ongoing humanitarian and healthcare missions to West Africa, imported LF cases will continue to challenge healthcare systems in high-income countries. The experience with these cases emphasizes the need for enhanced clinical awareness, early diagnostic capabilities, streamlined access to antiviral treatment, and focused public health responses targeting high-risk contacts rather than comprehensive community surveillance [100,101]. All results are continuously updated on the interactive Power BI dashboard at: https://safeinpregnancy.org/living-systematic-review-lassa/.

In conclusion, although several LF vaccine candidates demonstrate promising safety and immunogenicity profiles in healthy adults, critical evidence gaps remain for pregnant persons, children, and adolescents. This LSR serves as an essential tool for continuous monitoring, synthesizing, and rapidly disseminating emerging evidence, thereby supporting the timely advancement of vaccine development. Ultimately, this ongoing effort aims to contribute to the assessment of vaccine safety and effectiveness when vaccines are authorized to mitigate the devastating impact of Lassa fever, especially among the most at-risk populations in West Africa, by ensuring that public health decisions are informed by the best and most current available evidence.

## Supporting information

**S1 File. Supporting information (all supplementary figures and tables are included in this single file).**
(DOCX)

**S2 Table. PRISMA 2020 Checklist.**
(DOCX)

**S3 File. Database in comma-separated values.**
(CSV)

**S4 File. Dictionary of the database.**
(XLSX)

## Acknowledgments

The authors thank the administrative and technical staff at IECS and collaborating institutions for their support. We also thank the SPEAC members for their input and collaboration.

## Author contributions

**Conceptualization:** Ariel Bardach, Mabel Berrueta, Agustín Ciapponi, Edward P K Parker, Flor M Munoz, Pierre Buekens.

**Data curation:** Agustín Ciapponi, Juan M Sambade, Noelia Castellana, Jamile Ballivian, Martín Brizuela, Julieta Caravario, Daniel Comande, Esteban Couto, Agustina Mazzoni, Vanesa Ortega, Edward P K Parker, Florencia Salva, Katharina Stegelmann, Xu Xiong.

**Formal analysis:** Ariel Bardach, Mabel Berrueta, Agustín Ciapponi, Noelia Castellana, Jamile Ballivian, Martín Brizuela, Esteban Couto, Florencia Salva.

**Funding acquisition:** Mabel Berrueta, Agustín Ciapponi, Flor M Munoz, Pierre Buekens.

**Investigation:** Ariel Bardach, Mabel Berrueta, Agustín Ciapponi, Juan M Sambade, Jamile Ballivian, Julieta Caravario, Daniel Comande, Esteban Couto, Agustina Mazzoni, Vanesa Ortega, Edward P K Parker, Florencia Salva, Katharina Stegelmann, John S Schieffelin, Andy Stergachis, Flor M Munoz.

**Methodology:** Mabel Berrueta, Jamile Ballivian, Martín Brizuela, Julieta Caravario, Esteban Couto, Agustina Mazzoni, Edward P K Parker, Andy Stergachis, Flor M Munoz, Pierre Buekens.

**Project administration:** Agustín Ciapponi.

**Resources:** Mabel Berrueta, Agustín Ciapponi, Andy Stergachis.

**Software:** Juan M Sambade, Noelia Castellana, Flor M Munoz.

**Supervision:** Ariel Bardach, Mabel Berrueta, Agustín Ciapponi, Juan M Sambade, Jamile Ballivian, Agustina Mazzoni, Edward P K Parker, Flor M Munoz, Pierre Buekens.

**Validation:** Juan M Sambade, Martín Brizuela, Xu Xiong, Andy Stergachis, Pierre Buekens.

**Visualization:** Agustín Ciapponi, Edward P K Parker.

**Writing – original draft:** Ariel Bardach, Mabel Berrueta, Agustín Ciapponi, Juan M Sambade, Noelia Castellana, Jamile Ballivian, Julieta Caravario, Edward P K Parker, Flor M Munoz, Pierre Buekens.

**Writing – review & editing:** Ariel Bardach, Mabel Berrueta, Agustín Ciapponi, Juan M Sambade, Noelia Castellana, Jamile Ballivian, Martín Brizuela, Julieta Caravario, Daniel Comande, Esteban Couto, Agustina Mazzoni, Vanesa Ortega, Edward P K Parker, Florencia Salva, Katharina Stegelmann, John S Schieffelin, Xu Xiong, Andy Stergachis, Flor M Munoz, Pierre Buekens.

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
