## [Decision Letter · Decision Letter 0]

7 Oct 2025

Dear Dr. Bardach,

We look forward to receiving your revised manuscript.

Kind regards,

Osamudiamen Cyril Obasuyi, MD, MSc, FWACS, FMCOPh

Academic Editor

PLOS ONE

 [This work was supported by the Safety Platform for Emergency Vaccines (SPEAC), a Brighton Collaboration project funded by the Coalition for Epidemic Preparedness Innovations (CEPI). The funder (CEPI) had no role in study design, data collection and analysis, decision to publish, or preparation of the manuscript. The views expressed are those of the authors and do not necessarily reflect the positions of their institutions or the funder.]. 

Additional Editor Comments:

Dear Author

we thank you for submitting your research for peer review. While we find the research paper presents a topic which is very relevant to public health, there are however some technical points which needs to be addressed to enhance its suitability for publishing. In addition to the comments raised by the reviewers, please find below some additional points to address:

PRISMA CHECKLIST: **NO PRISMA CHECKLIST FOR ABSTRACTS INCLUDED!!!!** !!!!

A. Introduction

a. No explicitly stated objective of the SR

B. METHODS

a. No explicitly stated exclusion criteria

b. No mention of the searched databases

c. Methods used to decide study inclusion and exclusion are not explicitly stated.

d. Methods used to collect and synthesize results are not explicitly stated.

e. Data outcomes and suitability for analysis were not explicitly defined or stated.

f. All variables for which data were sought were not explicitly stated

g. Outcome measures were not specifically specified

h. How missing data was handled was not explicitly state

Reviewers' comments:

Reviewer's Responses to Questions

**Comments to the Author**

1. Is the manuscript technically sound, and do the data support the conclusions?

Reviewer #1: Yes

Reviewer #2: Partly

2. Has the statistical analysis been performed appropriately and rigorously?

Reviewer #1: N/A

Reviewer #2: Yes

3. Have the authors made all data underlying the findings in their manuscript fully available?

Reviewer #1: Yes

Reviewer #2: Yes

4. Is the manuscript presented in an intelligible fashion and written in standard English?

Reviewer #1: Yes

Reviewer #2: Yes

Reviewer #1: This review addresses Lassa fever vaccine development, focusing on safety, efficacy, and immunogenicity. Conducted as a living systematic review, it included 51 studies (2 clinical trials in adults, 49 preclinical). The clinical candidates (MV-LASV and rVSVΔG-LASV-GPC) demonstrated high immunogenicity (~95% seroconversion) and good tolerability, with no serious vaccine-related adverse events. Data are lacking for pregnant persons, children, and adolescents. Preclinical studies identified promising platforms for future clinical evaluation, highlighting the need for further trials in vulnerable populations.

The review systematically collects, evaluates, and synthesizes all available studies on a Lassa Fever vaccine. The topic is highly relevant to the field, and the review provides broad and up-to-date literature coverage. The manuscript is logically structured, coherent, and clearly presented, with claims and data accurately reported and well supported. To enhance transparency, please include the bibliographic references for the 51 studies included in the review (see Figure 1, Flowchart of Study Selection) and in Table 2.

The manuscript demonstrates a transparent and reproducible methodology, with a solid methodological evaluation. Its strengths lie in the originality of the contribution, which follows the authors’ previously published protocol, and in its overall clarity and readability.

The manuscript also addresses reactogenicity and tolerability, which are not reflected in the title. Furthermore, it may be clearer to discuss safety first, followed by immunogenicity and efficacy. Suggestions for improvement: consider revising the title to better reflect these key aspects in line with established vaccinology principles.

Reviewer #2: Dear authors,

You manuscript presents important findings and is of relevance for this journal. However there are some points that needs clarifications and adjustments.

1. Abstract

The abstract effectively communicates the importance of the study but currently exceeds the journal limit of 300 words (presently 335). This section should be shortened, with greater emphasis placed on quantitative findings and less on methodological description. For instance, details on the frequency of searches and use of software tools can be summarized more briefly, while the main results, such as the number of studies identified, the total number of participants in clinical trials, key immunogenicity outcomes, and gaps in special populations, should take priority.

The conclusions in the abstract should also adopt a more cautious tone, acknowledging the small sample sizes and preliminary nature of the data. Streamlining the abstract will improve readability and compliance with journal standards while ensuring readers immediately grasp the most important contributions of the study.

2. Introduction

While the manuscript cites surveillance and burden estimates, it would benefit from the integration of the most up-to-date figures from Nigeria, Liberia, and other endemic countries, especially given that recent outbreaks have generated new estimates of incidence and case fatality. Some references are dated and should be updated with newer analyses.

Furthermore, the authors mention including both clinical and preclinical studies; while this choice is reasonable, the justification should be more explicit.

PLOS ONE welcomes comprehensive syntheses, but the authors should clarify why combining both levels of evidence in one review adds value and how readers should interpret the relative weight of preclinical versus clinical data. Finally, the introduction could briefly mention the role of international funding and consortia (e.g., CEPI, WHO R&D Blueprint) in accelerating vaccine development, as this contextualizes the significance of the vaccine pipeline beyond scientific progress alone.

3. Methods

The inclusion and exclusion criteria for preclinical studies require further explanation. It is not clear whether these were restricted to peer-reviewed publications or whether grey literature and conference proceedings were systematically considered. Given that vaccine development often progresses with unpublished or non-peer-reviewed data, readers need to understand how the authors managed potential variability in study quality.

The handling of unpublished or abstract-only clinical data should be clarified. For example, one included trial is available only as a conference abstract/poster. The manuscript should explicitly explain how such evidence was evaluated in risk-of-bias assessments and how its lower certainty was reflected in the GRADE process.

The description of heterogeneity assessment could be expanded. While the authors note that I² statistics were used, the criteria for subgroup analysis, the thresholds for substantial heterogeneity, and the sensitivity analyses undertaken are not fully detailed. Explicitly describing how heterogeneity influenced the decision to pool or not pool studies would strengthen methodological transparency. Also, more information is needed on how missing outcome data were handled, especially given the small number of clinical trials and the variability in outcome definitions.

While the ethics statement is correctly marked as “N/A,” the methods should briefly clarify that all preclinical animal studies included in the review were published with stated ethical approvals when such information was available. Although the authors did not conduct the experiments, this ensures alignment with good practice in systematic reviews of animal data.

4. Results

The definition of immunogenicity outcomes, particularly “seroconversion,” should be made explicit. The included trials did not always provide consistent definitions, and without clarification, readers may struggle to interpret pooled findings. A short explanation of how the authors standardized or interpreted such outcomes would strengthen this section.

Safety outcomes are currently spread across the narrative, figures, and supplementary materials. Although the detail is valuable, the presentation could be simplified by including a consolidated summary table of adverse event frequencies, stratified by vaccine candidate. This would make the data far more accessible to readers who may not have technical expertise in interpreting forest plots.

The landscape tables mapping clinical and preclinical candidates are extremely detailed and important, but their current length interrupts the flow of the results. Key highlights should remain in the main text, while more detailed versions could be moved to the supplementary appendix. This would streamline the manuscript while ensuring full transparency through supplementary files.

While the description of preclinical evidence is helpful, the authors should explicitly remind readers that these data were not meta-analyzed and are not directly comparable to clinical findings. A clearer separation between descriptive mapping and quantitative synthesis would strengthen the results section.

5. Discussion

The interpretation of immunogenicity data—such as “95% seroconversion”—should be presented more tentatively. These results derive from a very small sample size (fewer than 90 participants across trials), and the uncertainty must be emphasized to avoid overstating robustness. Similarly, while the discussion on Lassa virus genetic diversity is important, statements about the potential for reduced cross-lineage protection remain speculative. I suggest that the authors should reframe these points as hypotheses requiring empirical validation rather than established conclusions.

The manuscript would benefit from a more explicit discussion of equity and ethics. It is striking that pregnant women, children, and adolescents remain systematically excluded from clinical trials. While the authors highlight this gap, they could go further in discussing the global health implications of continued exclusion of vulnerable groups. Addressing the ethical imperative for inclusion, as well as the structural challenges (e.g., regulatory hesitancy, limited trial capacity in endemic regions), would situate the review in a broader policy context.

The reliance on immunogenicity markers as surrogate outcomes must be discussed with greater nuance. Given that correlates of protection for Lassa fever are not well established, the predictive value of antibody titers is uncertain. The discussion should highlight this limitation clearly, especially since policy and funding decisions may hinge on such evidence.

The discussion could be enriched by reflecting on the practical strengths and challenges of conducting a living systematic review. The continuous update model is a clear strength, but it is also resource-intensive and dependent on timely trial reporting. Acknowledging these challenges would improve transparency and provide guidance for other groups considering similar approaches.

6. Figures and Tables

I suggest that all supplementary files and the interactive dashboard should be permanently archived in repositories that provide DOIs, such as Zenodo, OSF, or figshare. This is essential for reproducibility and long-term access, particularly given the living nature of the review. Without stable repositories, there is a risk that readers in the future may not be able to access key supplementary materials.

7. Language and Style

This is only a suggestion, not mandatory. The manuscript is written in clear English, but some sections are overly long and descriptive. Streamlining the language, especially in the abstract and discussion, would improve readability. Several sentences could be shortened without losing nuance. Redundancy should be avoided, particularly in the repeated explanation of Lassa fever epidemiology across the introduction and discussion. Careful editing will ensure the narrative remains concise and engaging.

8. References

The reference list is generally comprehensive and up to date, with inclusion of literature through April 2025. However, there are some technical issues. Duplicated references should be removed, and all URLs should be verified to ensure that they are live and accessible at the time of publication.

**Do you want your identity to be public for this peer review?** For information about this choice, including consent withdrawal, please see our Privacy Policy

Reviewer #1: **Yes: ** Manuela Chiavarini

Reviewer #2: No

---

## [Author Response · Author response to Decision Letter 1]

17 Oct 2025

Response to Academic Editor and Reviewers

Manuscript ID: PONE-D-25-48423

Title: Efficacy, safety, and immunogenicity of Lassa fever vaccines: a living systematic review and landscape analysis of vaccine candidates

Due date for resubmission: November 21, 2025 11:59 PM (America/Argentina/Buenos_Aires)

We thank the Academic Editor and reviewers for their constructive comments. Below we provide a point-by-point response. Reviewer comments are in italics; our responses follow, with page/line references to the revised manuscript.

Editor: Additional requirements and technical points

PRISMA for Abstracts checklist: We have added a PRISMA for Abstracts checklist as a separate file and ensured the abstract complies with itemized reporting.

Explicit SR objective: Added at the end of the Introduction (Objective paragraph).

Methods transparency: Methods now explicitly report databases searched, inclusion/exclusion criteria, study selection, data items, synthesis/heterogeneity handling, missing data, and ethics for animal studies.

Outcomes and variables: Data items and outcomes are now explicitly listed, including operational definition of seroconversion.

Figure captions: We have ensured separate figure legends for all figures in the manuscript.

Reviewer #1

Comment: Provide bibliographic references for all 51 included studies.

Response: Done. We cite each included study in Tables and in the References

Comment: Title reflects reactogenicity/tolerability and order of outcomes

Response: We retained the main title per PLOS ONE guidance but revised the abstract/results to present safety before immunogenicity and clarified tolerability/reactogenicity.

Reviewer #2

Comment: Abstract length and focus

Response: Abstract reduced to ≤300 words with emphasis on quantitative results and cautious conclusions.

Comment: Introduction updates and justification for including preclinical data

Response: We updated epidemiology references and added a brief justification for combining clinical and preclinical evidence and guidance on interpretation.

Comment: Methods—criteria for preclinical evidence, handling unpublished/abstract-only data, heterogeneity and missing data, animal ethics

Response: All topics clarified as requested; GRADE considerations for lower‑certainty abstract‑only data are described.

Comment: Results—definition of seroconversion; AE summary table; moving long landscape tables to Supplement; separate mapping vs synthesis

Response: Seroconversion definition added; we created a consolidated AE summary table in the Supplement; detailed landscape tables moved to Supplement with highlights in main text; mapping vs synthesis distinctions emphasized. We inserted abridged tables instead in the main paper.

Comment: Discussion—cautious interpretation, equity/ethics, correlates of protection, and LSR practicalities

Response: We tempered statements, expanded equity/ethics discussion, highlighted uncertainty around correlates, and reflected on strengths/challenges of an LSR.

Comment: Archiving of supplementary materials with DOIs

Response: We are prepared to archive the Supplement and dashboard exports in Zenodo/OSF with DOIs upon acceptance; we added this note to the Data Availability statement.

Comment: Language and style

Response: We streamlined the abstract and discussion to reduce redundancy and improve readability.

Comment: References—duplicates/URLs

Response: We screened for duplicates and verified URLs where applicable.

On behalf of all co-authors,

Ariel Bardach

---

## [Decision Letter · Decision Letter 1]

18 Nov 2025

Efficacy, safety, and immunogenicity of Lassa fever vaccines: a living systematic review and landscape analysis of vaccine candidates

PONE-D-25-48423R1

Dear Dr. Bardach,

We’re pleased to inform you that your manuscript has been judged scientifically suitable for publication and will be formally accepted for publication once it meets all outstanding technical requirements.

Kind regards,

Osamudiamen Cyril Obasuyi, MD, MSc, FWACS, FMCOPh

Academic Editor

PLOS ONE

Additional Editor Comments (optional):

Reviewers' comments:

Reviewer's Responses to Questions

**Comments to the Author**

Reviewer #1: All comments have been addressed

2. Is the manuscript technically sound, and do the data support the conclusions?

Reviewer #1: Yes

3. Has the statistical analysis been performed appropriately and rigorously?

Reviewer #1: Yes

4. Have the authors made all data underlying the findings in their manuscript fully available?

Reviewer #1: Yes

5. Is the manuscript presented in an intelligible fashion and written in standard English?

Reviewer #1: Yes

Reviewer #1: (No Response)

**Do you want your identity to be public for this peer review?** For information about this choice, including consent withdrawal, please see our Privacy Policy

Reviewer #1: **Yes: ** Manuela Chiavarini

---

## [Editor Report · Acceptance letter]

PONE-D-25-48423R1

PLOS ONE

Dear Dr. Bardach,

I'm pleased to inform you that your manuscript has been deemed suitable for publication in PLOS ONE. Congratulations! Your manuscript is now being handed over to our production team.

Kind regards,

on behalf of

Dr. Osamudiamen Cyril Obasuyi

Academic Editor

PLOS ONE